# Health, Performance Ratings and Approachability of 50–60-Year-Old Sedentary Adults (ActIv-Study): Key Insights for Health Economy and Exercise Promotion

**DOI:** 10.3390/ijerph21080969

**Published:** 2024-07-24

**Authors:** Dieter Leyk, Nadine Hartmann, Emanuel Vits, Thomas Rüther, Stefanie Klatt, Ralf Lämmel, Herbert Löllgen

**Affiliations:** 1Research Group Exercise Epidemiology, German Sport University Cologne, 50933 Cologne, Germany; leistungsepidemiologie@dshs-koeln.de (N.H.); emanuelvits@bundeswehr.org (E.V.); ruether@dshs-koeln.de (T.R.); herbert.loellgen@gmx.de (H.L.); 2Faculty of Computer Science, University Koblenz, 56070 Koblenz, Germany; laemmel@uni-koblenz.de; 3Department X-Anesthesia, Bundeswehr Central Hospital Koblenz, 56072 Koblenz, Germany; 4Institute of Exercise Training and Sport Informatics, German Sport University Cologne, 50933 Cologne, Germany; s.klatt@dshs-koeln.de; 5Faculty of Medicine, University Mainz, 55131 Mainz, Germany

**Keywords:** lifestyle, physical inactivity, health risk factors, barriers to sport, motives for sport, exercise promotion, incentive systems

## Abstract

Despite significant prevention efforts, the numbers of physically inactive individuals, chronic illnesses, exhaustion syndromes and sick leaves are increasing. A still unresolved problem with exercise promotion is the low participation of sedentary persons. This collective term covers heterogeneous subgroups. Their engagement with movement campaigns and resistance to change are influenced by numerous factors. Our aim was to analyse survey data on health, performance, lifestyle habits and the approachability to physical activity campaigns obtained from the Germany-wide ActIv survey. From 2888 study participants aged 50–60 years, 668 persons were categorised into the subgroups “never-athletes”, “sports-dropouts”, “always-athletes” and “sports-beginners”. Large and significant group differences were found for BMI, assessment of quality of life, health and fitness, risk factors and health problems. In total, 42.5% of “never-athletes” and 32.5% of “sports-dropouts” did not state any barriers to sport. There are substantial disparities between the non-athlete groups in terms of their motivation to exercise. In contrast, there are comparatively minor differences in motivation between “sports-dropouts” and “sports-beginners”, whose health and fitness are the primary motivators for sport. Our analyses suggest that (i) negative health and performance trends cannot be compensated for by appeals for voluntary participation in exercise programmes and (ii) powerful incentive systems are required.

## 1. Introduction

The enormous technological progress since the 1950s has led to huge changes in the living environments of modern societies, profoundly altering the lifestyles of nearly all demographic groups [1,2,3,4,5,6]. On the one hand, increasing prosperity has resulted in the availability of cheap and abundant food [7,8,9,10]. On the other hand, advancements in technology at work, in transportation, at home, and during leisure time have led to a massive reduction in physical activity [1,3,11,12,13,14].

Using data from the “U.S. National Health and Nutrition Examination Survey”, Church et al. [3] calculated the changes in energy expenditure and obesity prevalence of US employees from 1960 to 2008. In the 1960s, approximately 50% of the working population performed activities meeting the WHO recommendations for daily energy expenditure [15]. By 2008, this figure decreased to only 20%. Over this period, the average body weight of US employees increased from 76.9 kg to 91.8 kg. The globally visible consequences of hypercaloric diets and permanent sitting are known to be responsible factors for the significant increases in overweight, obesity, fatty liver disease, diabetes mellitus, circulatory disorders and other diseases [8,14,16,17,18,19,20].

Since 2008, the issue has become even more pressing, marked by a notable surge in mechanisation, automation and digitalisation in the workplace [21,22,23,24]. The spread of working from home and the future use of artificial intelligence may further increase the extent of physical inactivity among the majority of employees [5,6,25,26]. Changes in the household, leisure activities and transportation also contribute to worsening this issue [1,11,12,14]. Intensive and excessive consumption of digital media during leisure time is now prevalent across all generations worldwide [25,26,27,28,29]. This trend also applies to prolonged sitting in cars and on public transport [11,14]. In Germany, for example, 40 per cent of commuters also use their cars even for distances less than five kilometres [30].

The somatic consequences of a lack of exercise have been the focus of numerous studies in the past [7,17,20,31,32,33,34,35,36]. However, the significant effects on mental well-being are also becoming increasingly apparent [26,37,38,39]. Globally, the prevalences of burnout, mental disorders and depressive symptoms are on the rise [27,28,39,40].

For decades, education and prevention campaigns have targeted the motivation of individuals to adopt a more physically active lifestyle and enabling them to establish sufficient exercise and healthy dietary habits into their daily routines [6,8,14,15,20,41,42]. However, it is widely acknowledged that their effectiveness is insufficient and that lasting lifestyle changes are rarely achieved despite countless health campaigns. Despite better knowledge, numerous sources of information and offers (medical consultations, state health education, health insurance campaigns, etc.), the opportunities for active leisure activities are seldom utilised and unhealthy diets are usually continued [6,31,43,44].

One reason for this frequent failure may be the lack of important key data on target group-orientated and sustainable health promotion. Analysing the literature and campaigns that have been carried out provide a surprising result: although there are numerous studies, surveys and health reports, there are only a few target group-specific key data on strategic and application-oriented health and performance promotion [31,35,45,46,47]. The 2019 German Diabetes Surveillance Report, for example, lists detailed key figures on diabetes incidence, diabetes costs and the effectiveness of treatment programmes, and differentiates these by age, sex and regional distribution using location and dispersion measures [48]. However, there is no important distinction between physically active individuals and those who do not exercise.

Regular exercise is one of the most effective and cost-effective measures for the prevention and treatment of many chronic diseases [6,23,49,50,51,52]. If physical activity is to be effectively promoted for the prevention and treatment of diseases, precise knowledge about barriers to exercise, attractors and motives for exercising, everyday habits and health attitudes etc. is essential. One way of obtaining such data that has rarely been utilised to date is to conduct targeted surveys of persons with different levels of physical activity. Few studies show that the distinction between sporty and inactive groups of individuals is not sufficient [34,45,46,47,50,53].

In this study, therefore, a further differentiation was made between two non-athlete groups (“never-athletes” and “sports-dropouts”) and two athlete groups (“sports-beginners” and “always-athletes”). Through comparing these sub-groups, our objective was to identify prevention-relevant differences and important information for health campaigns were to be determined. Key questions we sought to address included: What motivates physically inactive individuals to engage in sports? What are the most common barriers to exercise? What factors prompt newcomers to sport to initiate or resume participation? For this purpose, we analysed over 10,000 data records from the Germany-wide ActIv (Activate Individuals) survey, which collects detailed health and performance-related data from the population. However, to mitigate the influence of age, our analysis focused exclusively on individuals aged 50–60.

Our study focussed on sedentary adults compared to groups engaged in sports activities. The objectives were twofold: (i) to identify differences between these groups in the assessment of health, fitness and lifestyle habits, as well as motives and barriers to exercise, and (ii) to increase the data for the development of targeted exercise and health campaigns tailored to specific demographic groups.

## 2. Materials and Methods

### 2.1. Data Collection

In the ActIv project (https://www.dshs-koeln.de/forschungsgruppe-leistungsepidemiologie/forschung/activ-projekt/ (accessed on 25 April 2024)), health-maintaining and disease-causing factors are analysed in the context of life situations, everyday habits and attitudes. Databases are modularised online surveys; survey links are available on portals tailored to specific educational, occupational and interest-based categories (e.g., technicians, teachers, taxi drivers, long-distance lorry drivers, nutrition, health, exercise, and sports forums, universities). In addition, the data collection was performed in regional survey facilities (e.g., in health centres, doctors’ surgeries, rehabilitation facilities, public and commercial facilities (clubs, fitness studios, medical supply stores, hairdressers)). Immediately after the survey, participants receive feedback regarding their individual health resources and risk factors, aimed at enhancing awareness of the importance of personal lifestyle choices. The ActIv project has obtained ethical approval from the ethics committee of the German Sport University Cologne and has implemented a comprehensive data protection protocol.

### 2.2. Survey Content

The survey content encompasses socio-demographic and anthropometric data (educational status, year of birth, sex, height, weight) and inquiries of health, fitness and quality of life, health complaints, medical history and medication consumption, everyday activities and sports participation, sports biography, and motives and barriers related to sports. Likert scales with five-point response options were used to measure personal assessments. The body mass index (BMI) was determined from the reported data on body weight and height.

Only completed and quality-checked questionnaires were analysed. The questionnaire items encompass nominal, categorical and scaled responses. The entries were automatically plausibility-checked.

### 2.3. Survey Sample

From data obtained from 10,041 ActIv participants, 2888 data sets of individuals aged 50–60 years were selected based on their sporting status and its duration (Table 1).

Participants were divided into the following categories: “never-athletes”, “sports-dropouts”, “sports beginners” and “always-athletes”, based on their responses to the following two questions: (1) “How do you categorise yourself in terms of sport?” (Possible answers: “non-athlete/recreational” or “healthy athlete/active competitive athlete”). (2) “Are you currently active in sport?” (Answer options: “Yes”/”No”, “I am currently taking a break” (for health, professional, or private reasons)). Depending on the answer, the participants were given different follow-up questions: After answering the second question, non-athletes were asked, for example, “When did you stop doing sport?”. They had the option of indicating the number of years or selecting the response option “I have never participated in any sports”. Athletes, on the other hand, were questioned “When did you start doing sports?” and had the two corresponding answer options.

The following inclusion and exclusion criteria were applied: only data from individuals who had refrained from participating in sports for a minimum of 20 years or their entire lives (“never-athletes”), or those who had consistently engaged in sports activities (“always-athletes”), were considered. The two groups of never-athletes and always-athletes served as anchor values.

The groups of “sports-dropouts” and “sports-beginners” only included persons who had not taken part in any sport or taken part in sport for at least 1 year and no more than 5 years. This was intended to prevent distortions resulting from information provided by individuals who had only been active in sport for a few months, for example, or who had to take a break from sport for several months due to injury or illness. The sport status groups were homogeneous in terms of sex and education.

### 2.4. Statistics

The data were analysed using IBM© SPSS© Statistics 24.0 (Armonk, NY, USA). The descriptive statistics included the calculation of position and dispersion measures (arithmetic mean, standard deviation). The comparison of dichotomous or ordinal-scaled parameters was carried out using cross-tabulations and chi-square tests. The comparison of variables from two independent samples was carried out using the T-test. Two-factor analysis of variance and Scheffé post-hoc test were used to compare interval-scaled characteristics. The prerequisites were checked before calculating the mean comparisons. A probability of error of *p* < 0.05 as accepted as significant for all tests.

## 3. Results

From 2888 study participants aged 50–60 years, 668 persons (296 men, 372 women) could be assigned to the four sport status groups (“never-athletes”, “sports-dropouts”, “sports-beginners”, “always-athletes”). Table 2 lists the group mean values for age, height, weight, BMI and group sizes (numbers of men (above) and women (below)). The mean differences in body weight and BMI mean values are significant (*p* < 0.05). Figure 1 shows the group differences (<0.05) in the BMI classes of the male and female study participants: among the “never-athletes”, only 12.9% of men and 10.0% of women are of normal weight, compared to 22.6% of men and 35.4% of women among the “sports-dropouts”.

### 3.1. Assessment of Own Quality of Life, Health and Fitness

There are also significant differences between the sport status groups in the assessment of quality of life, personal health and fitness (Figure 2). While 83.1% (“always-athletes”) and 62.5% (“sports-beginners”) of the active sports groups rated their quality of life as high, 55.8% of “sports-dropouts” and 40.4% of “never-athletes” did so (*p* < 0.05).

Figure 3 shows the group differences for the statement “Overall, I feel healthy” (*p* < 0.05). This statement is not true for 46.9% of “never-athletes” and 33.4% of “sports-dropouts”. Among the “sports-beginners” and “always-athletes”, 14.0% and 7.3%, respectively, disagree with this statement.

The group differences (*p* < 0.05) are even clearer in the assessment of performance (“Overall, I feel able to perform”), which can be seen in Figure 4. In total, 54.3% of the “never-athletes” and 41.8% of the “sports-dropouts” disagree with this statement. In the active sports groups, the figures are 18.6% (“sports-beginners”) and 7.9% (“always-athletes”).

### 3.2. Unfavourable Lifestyle Habits and Health Problems

Participants were asked about unfavourable lifestyle habits using a selection list (“Which lifestyle habits are so unfavourable to you that they could affect your health?”). Multiple answers were permitted for the 10 options listed. Table 3 shows the frequencies of the individual unfavourable lifestyle habits. It is easy to see that the two inactive groups cite unfavourable lifestyle habits more frequently (*p* < 0.05). Lack of exercise is most frequently seen as a potential cause of health impairments among the “never-athletes” (69.3%) and the “sports-dropouts” (53.3%).

Three or more negative lifestyle habits were reported by 40% of “never-athletes”, 31.1% of “sports-dropouts”, 18.6% of “sports-beginners” and 10.4% of “always-athletes”. No unfavourable lifestyle habits were reported by 13.1% of “never-athletes”, 16.2% of “sports-dropouts”, 30.2% of “sports-beginners” and 37.3% of “always-athletes”.

Health problems are reported significantly more frequently in the two non-athlete groups. In total, 56.1% of “never-athletes”, 55.1% of “sports-dropouts”, 39.6% of “sports-beginners” and 29.6% of “always-athletes” report that they have had health problems in the last 12 months. Table 4 shows the frequency of health complaints in the last 12 months. Back pain, muscle pain and joint pain occur most frequently in all sports status groups. In the last 12 months 37.5% of “never-athletes”, 38.3% of “sports-dropouts”, 19.5% of “sports-beginners” and 12.1% of “always-athletes” had at least three health problems.

### 3.3. Motivation to Partcipate in Sport

As expected, there are clear differences between the four sport status groups in terms of both motivation and barriers to take part in sport.

Figure 5 shows the study participants’ assessment of the statement “I am motivated to do sport”. In total, 87.9% of “always-athletes”, 66.7% of “sports-beginners”, 49.2% of “sports-dropouts” and 16.5% of “never-athletes” stated that they are motivated to participate in sport (*p* < 0.05). In the non-athlete groups, it is striking that over 50% of “never-athletes” disagree with the statement, while only around 15% of “sports-dropouts” do (*p* < 0.05).

The individual motives for participating in sports are listed in Table 5. In response to the question “What motivates you or would motivate you to do sport?”, the two sports groups stated sports motivators significantly more often than the non-athletes (*p* < 0.05).

Concerning campaigns promoting physical activity, the strongest motivators are particularly important for “never-athletes”, “sports-dropouts” and “sports-beginners”: in addition to health, body weight and stress reduction, physical performance is also a relevant attractor. It is noteworthy that only 30.1% of “never-athletes”, but 66.0% of “sports-dropouts” cite “enjoyment of sport” as a motive.

The group of “sports-beginners” was asked the additional question “Why did you start doing sport?”. The five most common reasons for taking up sport are physical reasons (70.3%), physical fitness (60.2%), weight reduction/weight management (55.5%), stress reduction/balancing (53.9%) and enjoyment of sport (32.8%).

### 3.4. Barriers to Sport

Participants were also asked about barriers to sport using a five-point Likert scale (“What prevents you from doing (more) sport?”). As depicted in Table 6, individuals who are inactive in sport cite more frequent and more numerous obstacles (*p* < 0.05). Health reasons, lack of motivation and lack of time were the three most frequently cited obstacles for “never-athletes” and “sports-dropouts”. However, an important result regarding the barriers is not shown in Table 6: 42.5% of the “never-athletes” and 32.5% of the “sports-dropouts” did not specify any sports barrier.

In the case of “sports-beginners” and “always-athletes”, lack of time is the most frequently cited barrier to sport at over 60%.

## 4. Discussion

Negative everyday habits such as lack of exercise and a hypercaloric diet not only lead to widespread chronic diseases and escalating medical expenses but also inflict considerable financial damage in the workplace [54,55,56,57,58,59]. Well before the onset of obesity and chronic illnesses (e.g., diabetes, fatty liver disease, circulatory diseases), employees often experience reduced resilience, declining productivity and a significant drop in performance [33,44,57,60,61,62].

Despite countless prevention campaigns, the number of days of incapacity for work, fatigue syndromes and chronic illnesses is increasing [52,63,64,65,66]. It is obvious that their effectiveness is not sufficient and that they need to be better targeted at “sedentary persons”. The fact that in Germany, for example, the majority of the population is chronically ill and ¾ of adults do not achieve the recommended amount of health-effective exercise [67,68,69] leads to the conclusion that there are numerous and heterogeneous prevention target groups in the population. Attractors and motives for exercise, barriers, responsiveness/accessibility to exercise campaigns and willingness to change can differ significantly among individuals of the same age, sex, education, occupation and place of residence and are also influenced by other factors (e.g., personal environment, life situation, sports biography, health, well-being and attitudes) [45,53,64,70]. More detailed knowledge about sedentary individuals is necessary in order to understand why they do not take part in exercise programmes, for example, and how they can be encouraged to incorporate physical and sporting activities into their everyday lives in the long term [46,70,71,72,73,74].

Surveys have been repeatedly used to obtain starting points and key data for effective physical activity promotion. The prerequisites and limitations of surveys should of course be known: our ActIv online surveys also have methodological pitfalls that should not be underestimated. In addition to incorrect entries, there are fundamental questions regarding distortions due to subjective assessments, missing objective measurement results (such as for body weight, height and BMI) and the extent to which the results of our cross-sectional surveys are representative. A further methodological limitation is that cross-sectional studies cannot necessarily prove causal relationships. Despite these limitations, focussing on the primary criterion “sports status” with the formation of the four main groups (never-athletes, sports-dropouts, sports-beginners and always-athletes) provides important key data: the data from the “sports-dropouts” and “sports-beginners” are particularly interesting, as it is possible to find out, for example, why “sports-beginners” have (re)started taking part in sport or what the most common barriers are for “sports-dropouts”.

As can be deduced from the list of the numerous factors influencing physical activity behaviour (see above), it is likely that differentiation into sport status groups is not sufficient either. It is obvious that non-athletes have individual inclinations, motives and limitations: Consider, for example, a permanently sedentary 35-year-old IT programmer (with a preference for soft drinks and fast food), a smoking single parent 43-year-old office administrator, a 54-year-old taxi driver with type II diabetes or a 62-year-old obese pensioner with gonarthrosis. To reduce the influence of the age factor and the age-associated bias, only individuals aged 50–60 years were included in our analyses.

As expected, there were statistically significant differences between the sport status groups in body weight, BMI (Table 2) and in the proportion of overweight and obese individuals (Figure 1). However, the group differences (*p* < 0.05) in the assessment of their own quality of life, health and fitness were unexpectedly clear (Figure 2, Figure 3 and Figure 4). It is worrying for the healthcare system and the economy that only 1/5 of “never-athletes” and ¼ of “sports-dropouts” feel healthy. Almost 40% of non-athletes had at least three health problems in the last 12 months. The self-assessments of fitness are just as worrying: only one in six “never-athletes” and one in four “non-athletes” believe they are fit. In view of ageing workforces and the lack of healthy and high-performing employees, these figures emphasise the need for effective health and exercise campaigns [6,14,31,44,51,58,59,62,72,75,76].

In this context, our analyses point to three findings relevant to prevention. Firstly, it should be noted that both non-athlete groups are obviously aware of unfavourable lifestyle habits (Table 3): exercise behaviour was cited as the most common negative everyday habit that can affect health. Conversely, this indicates that many physically inactive individuals are not lacking in knowledge. Instead, the central problem for sedentary persons appears to be insufficient motivation [5,21,25,34,44,45,47,55,73,74,77,78,79,80].

This leads to the second prevention-relevant finding of our study, the large differences in motivation among the non-athlete groups (Figure 5). The “sports-dropouts” are much more motivated to exercise than the “never-athletes” (49.20% vs. 16.50%). In contrast, only 15.2% of “sports-dropouts” disagree with the statement “I am motivated to do sports”. The negative attitude of this group is therefore close to that of the “sports-beginners” (11.6%), while the majority of “never-athletes” (52.5%) state that they are not motivated to exercise.

In their cluster analyses, Vanden Auweele et al. [47] already showed large differences in the approachability of non-athletes and the mostly lacking accessibility for exercise campaigns. The authors believe that only a small proportion of the 35–65-year-old physically inactive study participants will voluntarily establish an active lifestyle. Leyk et al. [46] also found varying degrees of motivation among non-athletic groups of 20–29-year-old men. The survey results of Smeets et al. [74] support the great importance of motivation. The study shows that primarily motivated individuals, but not unmotivated ones, were willing to read information and feedback on their physical activities.

It is likely that the motivation to exercise decreases with increasing duration of physical inactivity. The comparison of “never-athletes”, “sports-dropouts” and “sports-beginners” (Table 5) suggests that the differences in motivation between the two non-athlete groups are greater than between “sports-dropouts” and “sports-beginners”. The simple question “Since when have you not done any sport?” could therefore be a predictor for the accessibility of non-athletes to exercise programmes and relevant in counselling sessions. Incidentally, the biggest differences between “never-athletes” and “sports-dropouts” are in the motive “physical fitness” (delta 35.9%) and the motive “enjoyment of sport” (delta 26.2%). The question “What motivates you or would motivate you to do sport?” could also be a predictor for the accessibility of sedentary persons.

The third prevention-relevant finding of our study results from the information on the existing barriers. Approximately one third of the “sports dropouts” and over 40% of the “never-athletes” do not state any barriers. Lack of time is one of the three most common barriers (Table 6). These findings match the results of a 1-year model study [44]: the aim here was to estimate how many employees in a department could be motivated to participate in a broad-based health and fitness campaign under nearly ideal conditions. A total of 1010 employees were provided access to numerous sports programmes, monthly expert lectures and individual health consultations during working hours. However, fewer than 50% of the workforce took part in the kick-off events. By the end of the study, the number of participants had dwindled to less than 20%. The study shows that non-athletes are hardly motivated to take part in sport despite the excellent framework conditions. A notable 37.3% of non-athletes stated no obstacles or only one obstacle (over 50%). Lack of time was the most frequently cited barrier to sports participation, exceeding over 50%.

As to be expected among 50–60-year-old “sports-beginners”, health is the most common reason for (re)starting sport at 70.3%. It is worth noting that physical fitness is the second most common attractor at 60%. The newcomers to sport exercise almost three times a week and, at 187 min per week, were above the recommended WHO minimum duration of 150 min [6,14,15]. Older non-athletes can therefore definitely establish an active lifestyle. Finally, in this context, attention is drawn to another encouraging study for ageing and inactive societies [19]: nationwide surveys of long-distance runners and analyses of over 900,000 half/marathon participants aged 20–80 reveal that over 25% of persons aged 50–69 years had only initiated sports participation within the last 5 years. This high proportion of “newcomers to sport” shows that older former “non-athletes” can even successfully complete a marathon within a few years through regular training. Similar to health, weight reduction/control and stress reduction/balancing, physical performance evidently emerges as one of the most prevalent motives for engaging in sports.

## 5. Conclusions

There are major differences in the population in terms of motivation and willingness to participate in fitness and health promotion measures. A small proportion of individuals who are inactive in sports succeed in doing so, either independently motivated or with the help of good prevention campaigns.

Our genome is geared towards exercise. The drastic reduction in physical activity at work, in transport, at home and in our leisure time has led to the emergence of lifestyle environments that cause illness—despite the prosperity of modern societies. The bottom line is that digital media and artificial intelligence will further increase sedentary behaviour. Low-movement embossing in childhood and adolescence could turn even more persons into non-athletes and never-exercisers in the future.

It is obvious that the negative health and performance trends cannot be compensated for by merely appealing for voluntary participation in exercise programmes. In addition to structural measures to promote physical activity across all areas of society, education and prevention programmes, powerful incentive systems are required to motivate sedentary individuals to “rethink” their habits. These programmes should also teach the necessary personal skills to establish a healthy and performance-enhancing active lifestyle.

## Figures and Tables

**Figure 1 ijerph-21-00969-f001:**
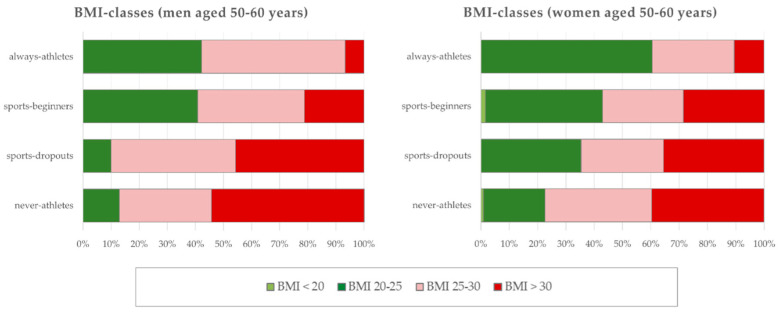
BMI classes of persons aged 50–60 years (296 men, 372 women) with different sport status (“always-athletes”, “sports-beginners”, “sports-dropouts”, “never-athletes”).

**Figure 2 ijerph-21-00969-f002:**
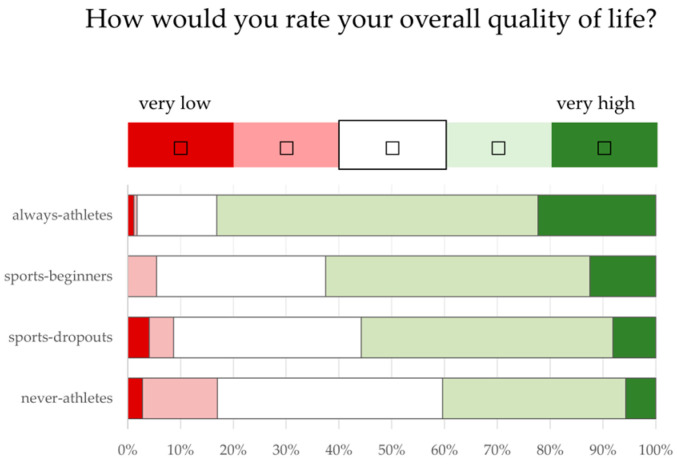
Assessment of quality of life (using a five-point Likert scale from very low (1) to very high (5). Responses from 50–60-year olds (n = 668) with different sporting statuses (“always-athletes”, “sports-beginners”, “sports-dropouts”, “never-athletes”) to the question “How would you rate your overall quality of life?”.

**Figure 3 ijerph-21-00969-f003:**
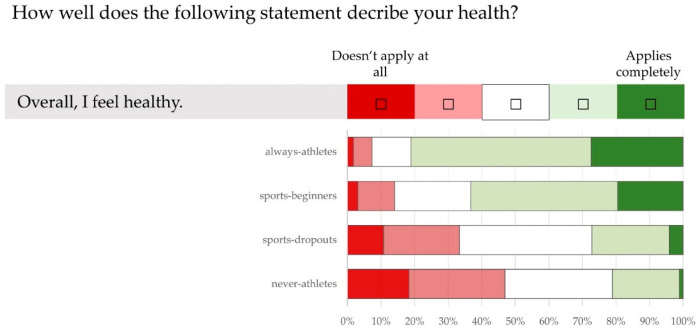
Assessment of health (using a five-point Likert scale). Information from 50–60-year olds (n = 668) with different sporting statuses (“always-athletes”, “sports-beginners”, “sports-dropouts”, “never-athletes”) on the statement “Overall, I feel healthy”.

**Figure 4 ijerph-21-00969-f004:**
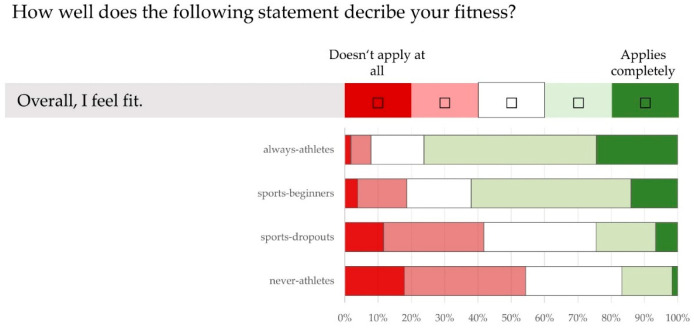
Assessment of own performance (using a five-point Likert scale). Information from 50–60 year olds (n = 668) with different sporting statuses (“always-athletes”, “sports-beginners”, “sports-dropouts”, “never-athletes”) on the statement “Overall, I feel able to perform”.

**Figure 5 ijerph-21-00969-f005:**
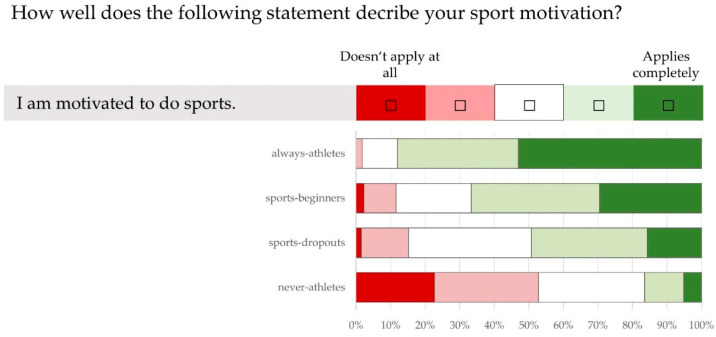
Assessment of motivation to take part in sport (using a five-point Likert scale)—responses from 50–60-year olds (n = 668) with different sporting statuses (“always-athletes”, “sports-beginners”, “sports-dropouts”, “never-athletes”) to the statement “I am motivated to do sport”.

**Table 1 ijerph-21-00969-t001:** Study samples of the four sport status groups selected from participants of the nationwide ActIv-project.

**16–99-year-old ActIv Participants (N = 10,041)**			
**50–60-year-old ActIv Participants (n = 2888)**			
**Study Sample of the Four Sport Status Groups (n = 668)**			
**never-athletes** (n = 176) no participation in sports ≥ 20 years	**sports-dropouts** (n = 197) sport drop out between 1 and 5 years	**sports-beginners** (n = 129) start in sport between 1 and 5 years	**always-athletes** (n = 166) participation in sports ≥ 20 years

**Table 2 ijerph-21-00969-t002:** Mean age, body height, body weight and body mass index (BMI) of 50–60-year-old men (above; n = 296) and women (below; n = 372) with different sport status (“always-athletes”, “sports-beginners”, “sports-dropouts”, “never-athletes”). Means + standard deviation.

Men	Never-Athletes	Sports-Dropouts	Sports-Beginners	Always-Athletes
Number of men	70	70	66	90
Age (years)	56.2 ± 2.4	55.8 ± 2.6	54.3 ± 2.8	55.2 ± 2.8
Body height (cm)	180.2 ± 7.3	181.2 ± 6.9	180.7 ± 6.0	179.3 ± 6.5
Body weight (kg)	101.3 ± 18.6	97.7 ± 15.6	86.4 ± 12.7	82.9 ± 10.2
BMI	31.16 ± 5.2	29.74 ± 4.4	26.46 ± 3.7	25.67 ± 2.8
**Women**	**Never-Athletes**	**Sports-Dropouts**	**Sports-Beginners**	**Always-Athletes**
Number of women	106	127	63	76
Age (years)	55.4 ± 2.7	55.6 ± 2.5	55.1 ± 2.7	55.0 ± 2.6
Body height (cm)	168.3 ± 7.3	167.2 ± 5.8	167.4 ± 5.1	168.4 ± 6.8
Body weight (kg)	85.7 ± 20.7	79.0 ± 16.9	78.0 ± 20.2	70.61 ± 13.4
BMI	30.5 ± 7.7	28.3 ± 5.7	27.8 ± 6.6	24.9 ± 4.2

**Table 3 ijerph-21-00969-t003:** Frequency of unfavourable lifestyle habits—responses from 50–60-year olds (n = 668) with different sporting status (“always-athletes”, “sports-beginners”, “sports-dropouts”, “never-athletes”) to the question “Which lifestyle habits are so unfavourable that they could affect your health?”.

	Never-Athletes	Sports-Dropouts	Sports-Beginners	Always-Athletes
Exercise behaviour	69.3%	53.3%	22.5%	11.4%
Diet/eating behaviour	39.2%	29.4%	25.6%	12.7%
Consumption of stimulants (coffee, sweets, etc.)	37.5%	46.2%	34.1%	36.1%
Sleeping habits	23.9%	28.4%	24.8%	18.1%
Smoking behaviour	22.7%	9.1%	11.6%	4.2%
Consumption of medication	9.7%	8.6%	3.1%	4.8%
Alcohol consumption	5.7%	7.1%	12.4%	16.9%
Social media/TV consumption/gaming habits	4.0%	7.1%	5.4%	4.8%
Drug use	0.0%	0.0%	1.6%	1.2%

**Table 4 ijerph-21-00969-t004:** Frequency of health problems in the last 12 months—responses from individuals aged 50–60 (n = 668) with different sporting statuses (“always-athletes”, “sports-beginners”, “sports-dropouts”, “never-athletes”) to the question “Have you experienced the following health problems in the last 12 months?”.

	Never-Athletes	Sports-Dropouts	Sports-Beginners	Always-Athletes
Back pain	76.10%	68.00%	51.20%	42.20%
Muscle or joint pain	73.90%	74.60%	57.40%	49.40%
Dizziness	31.30%	33.50%	21.70%	17.50%
Breathing problems	20.50%	19.80%	9.30%	4.20%
Chest pain	9.70%	9.60%	4.70%	4.20%
Fall due to health problems	6.80%	7.60%	6.20%	1.20%
Loss of consciousness	1.70%	1.50%	0.00%	2.40%

**Table 5 ijerph-21-00969-t005:** Most common motives for taking part in sport—responses from 50–60-year olds (n = 668) with different sporting statuses (“always-athletes”, “sports-beginners”, “sports-dropouts”, “never-athletes”) to the question “What motivates you or would motivate you to do sport?”.

	Never-Athletes	Sports-Dropouts	Sports-Beginners	Always-Athletes
Health reasons	69.3%	84.8%	88.4%	87.3%
Weight reduction/weight mangagement	61.4%	66.0%	69.8%	68.1%
Physical performance	49.4%	75.6%	79.1%	91.0%
Stress reduction/balance	41.5%	65.0%	62.0%	81.9%
Enjoyment of sport	30.1%	66.0%	54.3%	86.1%
Motivation to achieve something	26.1%	45.2%	47.3%	64.5%
Community experience	23.3%	41.1%	20.9%	41.6%
Sporty, good looks	22.7%	32.0%	25.6%	51.8%
Personal environment (acquaintances, friends, family)	18.8%	31.0%	24.8%	44.0%
Other reasons	3.4%	2.0%	3.1%	1.8%

**Table 6 ijerph-21-00969-t006:** Most common barriers to taking part in sport—responses from 50–60-year olds (n = 668) with different sporting statuses (“always-athletes”, “sports-beginners”, “sports-dropouts”, “never-athletes”) to the question “What prevents you from doing (more) sport?”.

	Never-Athletes	Sports-Dropouts	Sports-Beginners	Always-Athletes
Health reasons	70.9%	73.1%	36.5%	34.5%
Lack of motivation	53.4%	31.3%	25.0%	16.4%
Lack of time	45.6%	40.3%	63.5%	61.8%
No fun in sports	35.9%	9.7%	19.2%	5.5%
Lack of sports partners (acquaintances, friends, family)	35.9%	21.6%	11.5%	12.7%
Sport is too strenuous	31.1%	10.4%	17.3%	7.3%
No suitable sports programme	30.1%	24.6%	11.5%	5.5%
Other reasons	2.8%	7.6%	2.3%	1.2%

## Data Availability

The ActIv surveys are conducted using the Unipark programme developed by Tivian (https://www.unipark.com/ (accessed on 25 April 2024)). The data are stored on Tivian’s protected data servers and are not publicly accessible. Nevertheless, we are happy to answer questions about our data and analyses from colleagues, which can be sent to our email address: leistungsepidemiologie@dshs-koeln.de.

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
