# Peer review of "Health, Performance Ratings and Approachability of 50–60-Year-Old Sedentary Adults (ActIv-Study): Key Insights for Health Economy and Exercise Promotion"

_ijerph, 2024, doi:10.3390/ijerph21080969_

Round 1

Reviewer 1 Report

Comments and Suggestions for Authors

The purpose of this study is to examine measures to encourage participation in physical activity promotion campaigns. The analysis was based on the Germany-wide ActIv survey. Subjects aged 50-60 years were divided into groups according to their sporting activities and compared. The results of this study indicate that measures that appeal to voluntary motivation alone are unlikely to lead to participation in campaigns such as those promoting physical activity. I believe that this study is critical in order to construct measures to encourage participation in these campaigns.

The target population of this study was limited to the 50-60 years category. However, the reason for this limitation is not clearly stated. Limiting the target population is a very important part of clarifying the purpose of the study. In my opinion, it is necessary to clearly state why and for what purpose the age group of 50-60 years was chosen as the target. I would appreciate your consideration.

With regard to the Germany-wide ActIv survey on which this study is based, the data for the entire study population, 2,888 persons, is not mentioned at all in this paper. Please consider presenting all the data from the Germany-wide ActIv survey in tabular form. I would appreciate your consideration.

The author states that he utilized the two questions in the text of the paper as a way of grouping the study subjects (L138-145). However, there is no clear explanation of the relationship between the answers to the two questions and the four group classifications. The rationale for the group classification is a very important point that needs to be clarified, in my opinion. Please consider this.

With regard to quality of life, it is stated that the survey was conducted according to a 5-point Likert scale (L193-195); please specify in the paper what scale was used in the quality of life survey.

A clear definition of “sedentary persons,” as discussed in the paper, should be provided. Also, the relevance of “sedentary persons” to the four group categories is mentioned at the beginning of the discussion, but I think it is necessary to clearly state the relevance using previous studies. I would appreciate your consideration.

Author Response

We would like to thank Reviewer 1 for the constructive and helpful comments!

Comments and Suggestions for Authors

  1. The purpose of this study is to examine measures to encourage participation in physical activity promotion campaigns. The analysis was based on the Germany-wide ActIv survey. Subjects aged 50-60 years were divided into groups according to their sporting activities and compared. The results of this study indicate that measures that appeal to voluntary motivation alone are unlikely to lead to participation in campaigns such as those promoting physical activity. I believe that this study is critical in order to construct measures to encourage participation in these campaigns.

We would like to thank Reviewer 1 for the valuable comments and constructive suggestions. We are pleased that Reviewer 1 attaches great importance to our study for the planning of effective health campaigns. We firmly believe that more precise knowledge and basic data on lifestyle habits, motives and barriers to physical activity, including the sport status groups, are necessary to better reach target groups and individuals in the future, to offer more appropriate and efficient physical activity promotion and also to develop powerful incentives (bonus and malus systems).

  1. The target population of this study was limited to the 50-60 years category. However, the reason for this limitation is not clearly stated. Limiting the target population is a very important part of clarifying the purpose of the study. In my opinion, it is necessary to clearly state why and for what purpose the age group of 50-60 years was chosen as the target. I would appreciate your consideration.

It is undisputed that performance losses, health complaints and illnesses occur more frequently with increasing age. It can also be assumed that there are clear differences between individuals aged 35, 45 and 55 years, for example, when it comes to barriers to sport, sporting motives and life situations.

The focus on the target population and the reason for age limitation explained in the manuscript (see lines 315-320): “It is obvious that non-athletes have individual inclinations, motives and limitations: Consider, for example, a permanently sedentary 35-year-old IT programmer (with a preference for soft drinks and fast food), a smoking single parent 43-year-old office administrator, a 54-year-old taxi driver with type II diabetes or a 62-year-old obese pensioner with gonarthrosis. To reduce the influence of the age factor and the age-associated bias, only individuals aged 50-60 were included in our analyses.”

The limitation to 50-60 year old persons was intended to reduce age-related influences as well as age differences between the four comparison groups. This age range is also the particular focus of many company health promotion campaigns, as it is becoming more important for older employees to remain healthy, fit and resilient in the face of demographic change and labor shortages.

Age-related differences (e.g. comparison between age groups of 20-35yrs, 35-50 yrs, 50-65 yrs and over 65 yrs persons) will be analysed once a sufficiently large database is available.

  1. With regard to the Germany-wide ActIv survey on which this study is based, the data for the entire study population, 2,888 persons, is not mentioned at all in this paper. Please consider presenting all the data from the Germany-wide ActIv survey in tabular form. I would appreciate your consideration.

We followed this suggestion and included a corresponding table in the manuscript (see chapter ”2.3 Survey sample”)

  1. The author states that he utilized the two questions in the text of the paper as a way of grouping the study subjects (L138-145). However, there is no clear explanation of the relationship between the answers to the two questions and the four group classifications. The rationale for the group classification is a very important point that needs to be clarified, in my opinion. Please consider this.

The selection of never-athletes, sports-dropouts, sports-beginners and always-athletes from the 2,888-strong data set of 50-60 year olds was based on the two questions and was intended to ensure that only people who met the inclusion criteria for the respective groups were selected. The two questions (1. How do you categorise yourself in terms of sport? 2. Are you currently active in sport?) were filter questions. Depending on the answer, the participants were given different follow-up questions: After answering the 2nd question, non-athletes were asked, for example, "When did you stop doing sport?" with the answer options (i) "I've never participated in any sport" or (ii) “I haven't done any sport since xxxx (year)”. Athletes, on the other hand, were questioned "When did you start doing sport?" with the answer options (i) "I've always done sport" or (ii) "I've been doing sport since xxxx (year)". We have added further information on the filter questions to the manuscript (see chapter 2.3. Survey sample).

  1. With regard to quality of life, it is stated that the survey was conducted according to a 5-point Likert scale (L193-195); please specify in the paper what scale was used in the quality of life survey.

We used the question “How would you rate your overall quality of life” and a 5-point Likert scale. The response scale ranges from the anchor values very low (=1) to very high (=5). We added this information in the paper (see legend Figure 2)

Reviewer 2 Report

Comments and Suggestions for Authors

 The study presents conceptual issues that directly impact data collection and analysis, which cannot be adjusted since the data has already been collected. The theoretical basis for the division between “never-athletes,” “sports-dropouts,” “sports beginners,” and “always-athletes” is unclear. This division seems arbitrary and lacks foundation. Moreover, the title does not relate to this division and does not convey the essence of the study. The introduction is overly long and lacks coherence, often blending methods and discussion. The reduction from 10,041 to 2,888 participants is not justified, nor are the criteria for group separation. Additionally, basic errors such as the use of "gender" and "weight" are evident, along with a repetitive and tedious writing style that detracts from the main objectives.

Comments on the Quality of English Language

See my comments to the authors. 

Author Response

Reviewer 2

Comments and Suggestions for Authors

  1. The study presents conceptual issues that directly impact data collection and analysis, which cannot be adjusted since the data has already been collected. The theoretical basis for the division between “never-athletes,” “sports-dropouts,” “sports beginners,” and “always-athletes” is unclear. This division seems arbitrary and lacks foundation. Moreover, the title does not relate to this division and does not convey the essence of the study. The introduction is overly long and lacks coherence, often blending methods and discussion.

Thank you for your review and for providing feedback on our manuscript. We appreciate the time and effort you have taken to critique our work. While we regret that your assessment was negative, we acknowledge the importance of addressing your concerns to improve the quality of our study. Below, we provide our responses to the key points you raised.

Conceptual Issues and theoretical basis for the division between “never-athletes, sports-dropouts, sports beginners, and always-athletes”: We believe that there is a misunderstanding here. This classification is grounded in existing literature that demonstrates significant health, performance, and motivational differences among exercise groups. We use this approach successfully in other studies, too. However, we recognize that this was not sufficiently explained in the original manuscript and have clarified this in the revised version.

  1. The reduction from 10,041 to 2,888 participants is not justified, nor are the criteria for group separation.

The reduction from 10,041 to 2,888 participants was based on our objective to minimize age-related biases by including only individuals aged 50-60 in our analyses. We apologize for not making this sufficiently clear in the manuscript. In the revised version, we have included a table that justifies the selection criteria and the resultant sample sizes (see chapter 2.3. Survey sample). Moreover, we have added further information on the filter questions, too.

  1. Additionally, basic errors such as the use of "gender" and "weight" are evident, along with a repetitive and tedious writing style that detracts from the main objectives.

We regret that our writing style was perceived as repetitive and tedious. We have made efforts to enhance the readability and engagement of the text. Regarding the terminology, we have reviewed the use of terms such as "gender" and "weight" to ensure accuracy and appropriateness.

Round 2

Reviewer 2 Report

Comments and Suggestions for Authors

The responses provided by the authors are insufficient. In fact, the adjustments to my first point not only fail to help but worsen the situation regarding discrimination between the groups.

First, concerning the authors' response: "We believe that there is a misunderstanding here. This classification is grounded in existing literature that demonstrates significant health, performance, and motivational differences among exercise groups. We use this approach successfully in other studies, too. However, we recognize that this was not sufficiently explained in the original manuscript and have clarified this in the revised version."

Which literature? Which study classified and validated these discriminations? This was not presented, and yet the authors insist that this division is based on the literature. Regarding previous publications, it is unfortunate that this passed. The divisions are extremely subjective and lack scientific basis. For example, what classification of "athletes" are you using? Does the "athlete" compete at an amateur, regional, national, or international level in this division? If I was an amateur athlete, do I fall under this classification? And what about "always-athletes"? From what age do you consider elite athletes? If we compare football players and gymnasts, there is a huge difference in the age at which they start competing at a high level.

Finally, errors in nomenclature are still present.

In summary, the study is extremely weak and does not contribute to the current literature.

Author Response

Response to reviewer 2:

Once again, reviewer 2 criticises massively the
I. four sports activity groups and their "conceptual and theoretical basis"
II. limitation of the research collective to 50 to 60-year-olds
III. language and semantics of our manuscript.

We regret that reviewer 2 cannot comprehend our manuscript revision and our explanations. Even worse, in his eyes our revision even represents a deterioration. We do not share these assessments.

ad I)    Formation and justification of the sports activity groups
1) In the introduction, the importance of everyday, occupational and sport-related activities for the health status of the population is explained in detail. As a result, sport activity and individual sport status are inevitably of decisive importance for prevention and the conception of health-promoting measures.
2) In the literature reference (para. 76 ff.) it is derived from the study situation, among other things, that the "exercisers" and "non-exercisers" (Ref. 49 Vaden Auweele 1997) as well as "non-movers" (Ref. 45 Rütten et al. 2009, Ref. 53 Rütten et al. 2007) have already been the subject of research. However, there is still a lack of target group-specific key data.
3) Our own studies have also already focused on the health, fitness and lifestyle conditions as well as the motives and barriers of physically active and inactive people (Ref. 7 Leyk et al. 2008, Ref. 19 Leyk et al. 2010, Ref. 43 Leyk et al. 2012, Ref. 46 Leyk at al. 2012, Ref. 49 Leyk 2009, Ref. 79 Leyk et al. 2019).
In addition, we conducted a multi-month intervention study on health promotion and physical activation of employees in the workplace (Ref. 44 Leyk et al. 2014). One of the most important results was that, despite best conditions and high personnel costs, the prevention programmes were only accepted by the primary target group, physically inactive people, in isolated cases at best.
4) The aforementioned studies led to the ActIv study approach and the logically coherent, more complete group categorisation into "always-athletes and never-athletes" and the respective "sport beginners and sport leavers" groups. 
This approach was used to close the sports biographical gap that existed in previous studies between the "ever-athletes" and "non-athletes", which functioned as anchor points in the present study.
5) Our questionnaire design is based on the expectation and hypothesis that target group-specific patterns and profiles of lifestyle and health factors, motives, etc. can be recognised significantly better. The available study data on self-assessed quality of life, health, fitness, sport motives and sport inhibitions provide impressive evidence of these polarising patterns. The ActIv project can provide further insights in the future, especially as the surveys record other socio-demographic characteristics such as education, occupation and place of residence. These results will of course be published as the number of cases increases. 
6) Individual sport preferences, which are also recorded in the survey, can also be taken into account. In our manuscript, however, the sports disciplines and the differentiation into different performance levels are not relevant. The sport and exercise recommendations of the WHO are also not structured in such a detailed way as reviewer 2 would like.

ad II)    Limitation of the study cohort to 50 to 60-year-olds
7) The focus on 50-60 year olds is sufficiently justified in the manuscript in terms of case numbers, statistics and content and is at the scientific discretion of the authors. It is in our interest to present the characteristics of all age groups once a sufficient number of cases has been reached.
8) The health, performance and resilience of this age group is of great importance for companies and the healthcare system. The results of our study are highly relevant for target group-adjusted health promotion and prevention measures.
9) As mentioned above (see point 6), in the ActIv survey we not only record whether a person is a competitive athlete or a recreational athlete, but also which type of sport they do, how often and for how long they train and which type of sport they do. However, this data does not belong in this paper. The statements by the expert are unhelpful and surprising (e.g. in view of workplace promotion of physical activity of employees aged 50 to 60 years).

ad III)    Language and semantics of our manuscript
10) The reviewer criticises the choice of words "sex" or "gender", "weight" or "body mass" and sees fundamental errors here. We are well aware of the differences: Nobody will deny, for example, that "gender" addresses the sociological dimension of gender-related lifestyle assessments better than the biologically dominated gender definition "sex". This can be discussed, but it distracts attention from more important issues. We have reviewed the manuscript and made linguistic adjustments.

Although reviewer 2 continues to criticise our publication for reasons that are not clear to us, we are convinced that our preliminary descriptions are more than sufficient to demonstrate the high quality of our study methodology and our manuscript. The numerous enquiries from other publishers during the pre-publication phase have also confirmed our assessment.
